# Species Composition and Seasonal Abundance of Predatory Mites (Acari: Phytoseiidae) Inhabiting *Aesculus hippocastanum* (Sapindaceae)

**Michal Kopačka and Rostislav Zemek ***

Institute of Entomology, Biology Centre CAS, 37005 České Budějovice, Czech Republic
* Correspondence: rosta@entu.cas.cz

**Abstract:** Species of the family Phytoseiidae (Acari: Mesostigmata) inhabit a wide range of herbs, shrubs, and trees. Horse chestnut, *Aesculus hippocastanum*, is an important ornamental tree in Europe and, in addition to its importance for pollinators, it can serve as a host plant of these predatory mites. Little is, however, known about the factors affecting spatiotemporal variability in the density of phytoseiids on *A. hippocastanum* in an urban environment. We therefore assessed the species composition and the spatial and seasonal variability in the abundance of Phytoseiidae species in the city of České Budějovice, South Bohemia, Czech Republic. Compound-leaf samples were randomly collected from horse chestnut tree branches at eight sites, five times during the vegetation season in 2013. The mites were collected by washing technique and mounted on slides for identification. In total, 13,903 specimens of phytoseiid mites were found, and eight species were identified: *Amblyseius andersoni*, *Euseius finlandicus*, *Kampimodromus aberrans*, *Neoseiulella tiliarum*, *Phytoseius macropilis*, *Paraseiulus talbii*, *Paraseiulus triporus,* and *Typhlodromus (Typhlodromus) pyri*. The predominant species was *E. finlandicus* (96.25%). The mean number of mites per compound leaf was 2.53, 10.40, 23.54, 11.59, and 9.27 on the sampling dates in each month between May and September, respectively. The results further revealed that the mite abundance varied significantly among sampling sites and that it was negatively related to percentage of greenery area, intensity of greenery care, distance to water body, and density and age of horse chestnut trees, while it was positively related to air pollution index. The importance of leaf micromorphology for the attractiveness of *A. hippocastanum* to Phytoseiidae is discussed.

**Keywords:** horse chestnut tree; diversity; population dynamics; mite density; spatial variability; city parks





## 1. Introduction

The horse chestnut, *Aesculus hippocastanum* L. (Sapindales: Sapindaceae), is native to the mountains of the Balkans in southeast Europe. The biggest native populations are in mainland Greece (Thessaly and Pindus Mountains), but its native range also includes populations in Albania, the Republic of Macedonia, Serbia, and eastern Bulgaria [1]. This tree species has been planted in Europe since the seventeenth century primarily for ornamental purposes. Many *A. hippocastanum* trees are currently grown in city parks and urban forests for their beauty, providing shade and reducing the urban heat island effect. Moreover, extracts from the various parts of this tree have been widely used in herbal medicine [2]. The horse chestnut trees are also planted in forests in order to increase and diversify food resources because seeds are eaten by deer and wild boar [1], though the presence of various metabolites makes the seeds a less attractive food source [3].

*Aesculus hippocastanum* attracted researchers' attention when the horse chestnut leafminer, *Cameraria ohridella* Deschka and Dimic (Lepidoptera: Gracillariidae), became an invasive pest and spread to all countries of Europe [4–6]. Larvae of this moth inflict substantial leaf damage,

and together with the horse chestnut leaf blotch caused by *Guignardia aesculi* (Peck) V.B. Stewart (Botryosphaeriales: Botryosphaeriaceae), cause a decline in horse chestnut trees' aesthetical value [7]. Although ornamental function and providing shade are the main reasons why growing *A. hippocastanum* is so popular in city parks, other functions of this tree species might be less known. For example, horse chestnut provides pollen and nectar for pollinators [8]. *Aesculus hippocastanum* also serves as a habitat for many organisms, including natural enemies such as predatory mites of the family Phytoseiidae (Acari: Mesostigmata).

A survey of phytoseiid mites on deciduous trees and bushes conducted in Finland in 1989–1991 revealed the highest density of mites to be on *A. hippocastanum*, with 1063 mites per 100 leaves on average and a maximum of 14.4 mites per leaf in a single sample [9,10]. Similar results were reported for South Bohemia, Czech Republic, where the population density of phytoseiids ranged between 1 and 28, with a mean number of mites per compound leaf of 10.5 [11]. The predominant species in both Finland and in the Czech Republic was *Euseius finlandicus* (Oudemans), which represented more than 90% of all phytoseiids found [9,11,12]. In Greece, the number of phytoseiids was found to be lower, ranging between 0 and 16, with a mean density of 4.2 mites per compound leaf; another species, *Kampimodromus aberrans* (Oudemans), was nearly as abundant as *E. finlandicus* [11].

Although the above data provide basic information on mite abundance, the spatiotemporal variability in the density of phytoseiids on *A. hippocastanum* in an urban environment has not yet been studied. The objectives of the present study were therefore to investigate the species composition of phytoseiid mites inhabiting horse chestnut trees and seasonal changes in their abundance within a medium-sized city. Since various factors affect populations of both herbivorous and phytoseiid mites, as demonstrated, e.g., by Fidelis et al. [13] and Tixier et al. [14], we also analyzed which factors might have a significant effect on the abundance of Phytoseiidae. For example, we were interested as to whether greenery maintenance has any effect, as it was shown to in the case of horse chestnut leaf miner parasitoids [15], or if mite density is affected by air pollution [16].

## 2. Materials and Methods

### 2.1. Sampling Sites

The research was carried out in the town of České Budějovice, Czech Republic (48°59′ N, 14°29′ E; 386 m above sea level). The town, which is predominantly surrounded by forests and agricultural fields, has about 260 hectares of open public green space inhabiting 534 horse chestnut trees [17]. Eight sampling sites were defined within the cadastral area of the town (Figure 1), which differed in several parameters (Table 1).

**Table 1.** Characteristics of *Aesculus hippocastanum* sampling sites within České Budějovice.

| Site Name | Geographical Coordinates of Geometric Centre | Greenery | | Distance to Water [2] (m) | *A. hippocastanum* | |
|---|---|---|---|---|---|---|
| | | Area (%) | Care [1] | | Density (ha$^{-1}$) | Age [3] (years; x̄ ± SE) |
| City center | 48.9744136 N, 14.4770594 E | 9.02 | 3 | 69 | 1.40 | 74.99 ± 1.86 |
| Šumava and Máj estate | 48.9841631 N, 14.4407714 E | 31.69 | 2 | 660 | 0.22 | 39.02 ± 3.19 |
| Vltava estate | 48.9962106 N, 14.4519647 E | 36.96 | 2 | 360 | 0.43 | 41.79 ± 3.49 |
| Třebotovice and Kaliště village | 48.9612606 N, 14.5651828 E | 6.67 | 1 | 700 | 0.17 | 17. 82 ± 6.86 |
| Rožnov estate | 48.9601436 N, 14.4763944 E | 9.86 | 2 | 110 | 0.19 | 70.49 ± 4.43 |
| Pražské předměstí estate | 48.9860569 N, 14.4674681 E | 29.84 | 2 | 338 | 0.40 | 49.74 ± 2.75 |
| Stromovka park | 48.9670208 N, 14.4551158 E | 40.00 | 2 | 102 | 0.30 | 43.53 ± 4.13 |
| Nádražní street | 48.9779942 N, 14.4866511 E | 7.69 | 0 | 920 | 1.38 | 77.74 ± 3.02 |

[1] Intensity of greenery care: 0—no mowing or raking, 1—mowing 1–2 times per season, 2—mowing 3–4 times per season, 3—mowing >4 times per season. [2] The shortest distance from geometric center to larger water body (river, pond, or stream). [3] Tree age was estimated by the method proposed by Jura [18].

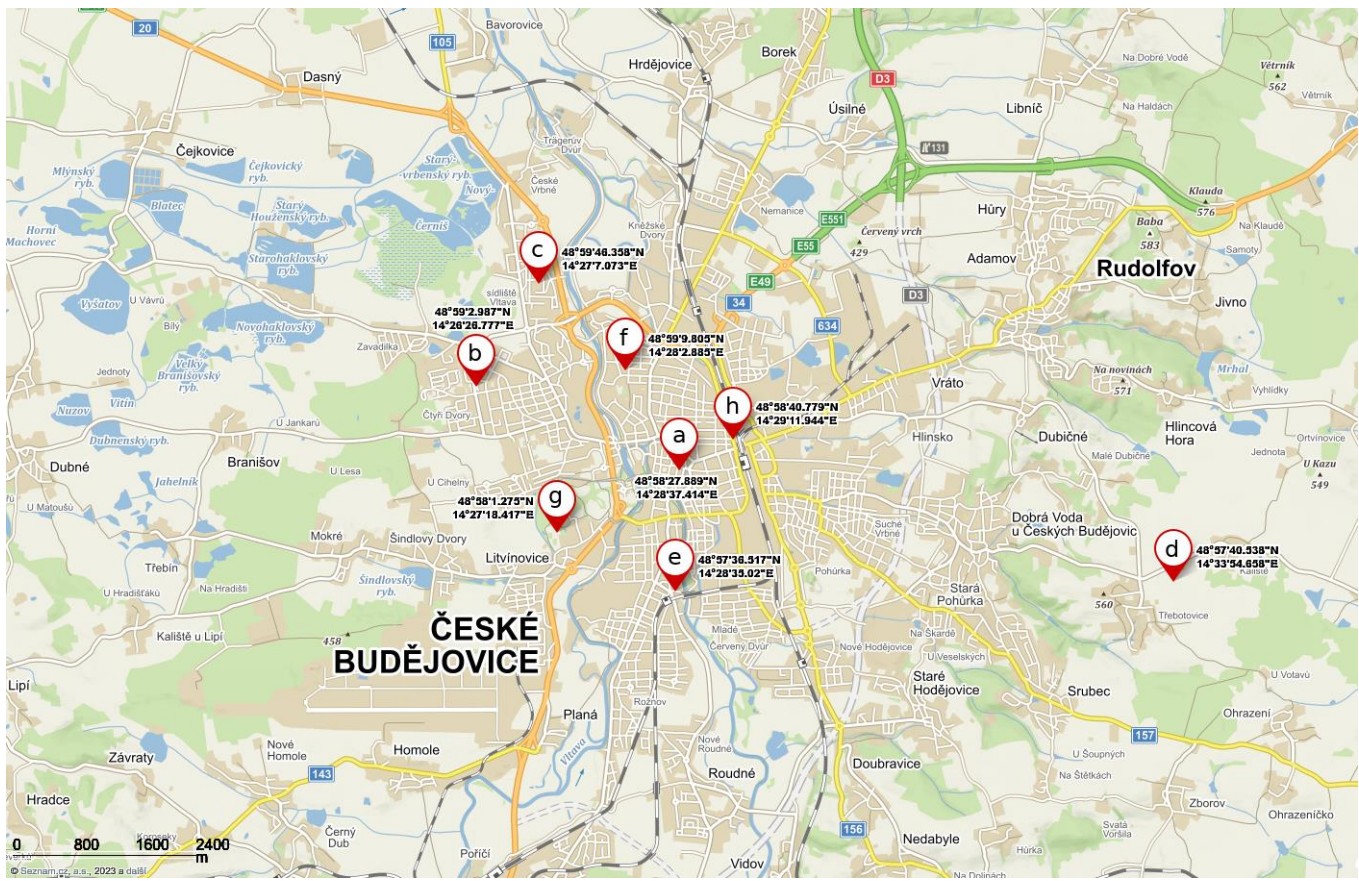

**Figure 1.** The map of České Budějovice with marked sites where *Aesculus hippocastanum* leaf samples were collected. (**a**) City center; (**b**) Šumava and Máj estate; (**c**) Vltava estate; (**d**) Třebotovice and Kaliště village; € Rožnov estate; (**f**) Pražské předměstí estate; (**g**) Stromovka park; (**h**) Nádražní street. Source: Mapy.cz.

Weather data are shown in Figure 2. The highest daily mean temperature and sum of precipitation were found in July and June, respectively (Table 2). In addition, data on air pollution provided by Czech Hydrometeorological Institute were obtained for each sampling site from Geographical Information System of South Bohemia [19] (Table S1).

**Table 2.** Mean temperature and sum of precipitations in České Budějovice for the period May–September 2013.

| Month | Temperature in °C (Mean ± SE) | Precipitations in mm |
|---|---|---|
| May | 12.95 ± 3.17 | 84.5 |
| June | 16.95 ± 4.96 | 190.0 |
| July | 20.35 ± 2.53 | 74.2 |
| August | 18.93 ± 3.77 | 59.5 |
| September | 13.80 ± 2.98 | 35.2 |

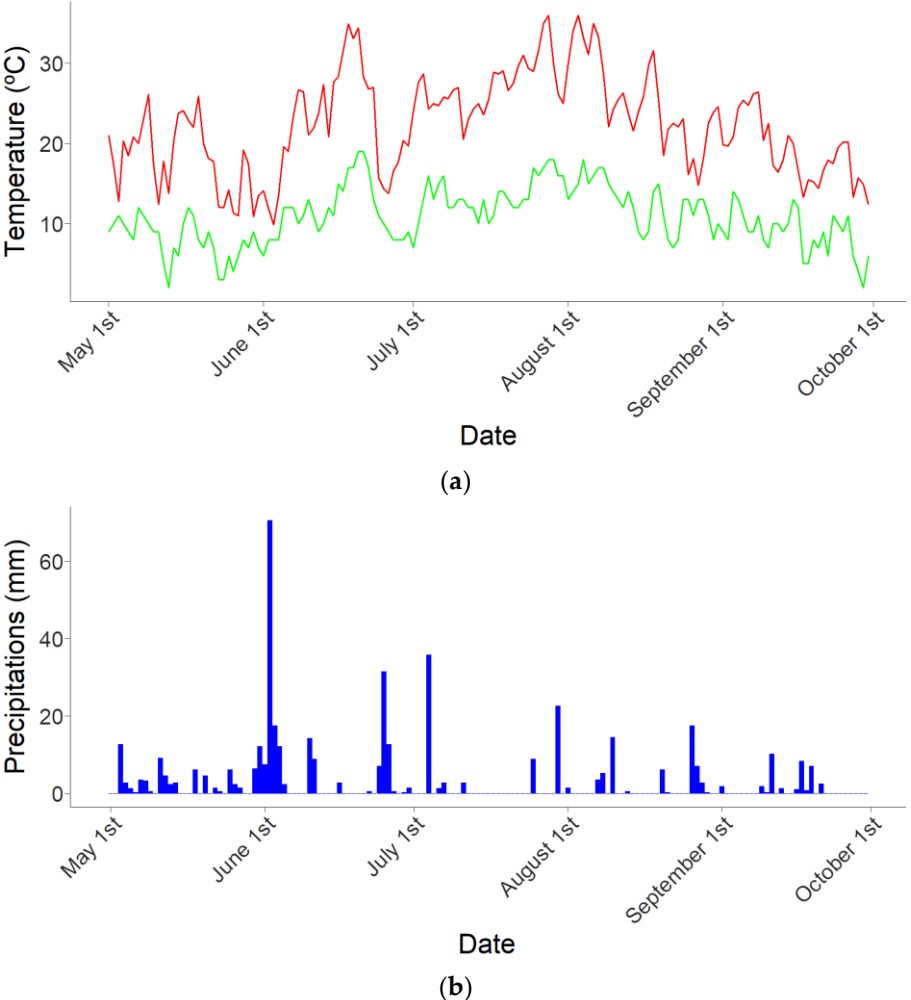

**Figure 2.** The maximum (red line) and minimum (green line) daily temperatures (**a**) and daily precipitations (**b**) in České Budějovice during sampling season in 2013.

## 2.2. Sampling of Horse Chestnut Leaves

The population of Phytoseiidae was assessed on horse chestnut leaves that were collected randomly five times during the vegetation season in 2013. The sampling dates were from 16 to 24 May, from 13 to 21 June, from 11 to 19 July, from 11 to 22 August, and from 9 to 19 September. The sampling took place only when there had been no rain for at least 48 h prior to sampling. A randomly selected compound leaf collected from *A. hippocastanum* tree up to 2.5 m above the ground represented the sample unit. In total, 30 compound leaves were collected per sampling site, each leaf from a different tree except at sites with fewer than 30 trees. The sampled trees were selected evenly across the whole site. The leaves at each individual site were all sampled on the same day, placed individually into polyethylene bags, and transported in a portable cool box to the laboratory where the leaves were stored at a temperature of 4 °C for a maximum of 24 h before they were processed.

## 2.3. Collection and Identification of Phytoseiid Mites

The mites were collected from individual leaves using the washing technique [20]. After taking the photographs, the horse chestnut leaf was held by the petiole, and individual leaflets were cut off by pruning shears into a glass jar (volume 700 mL) containing 350 mL of 85% ethanol. The jar was closed carefully and shaken vigorously for two minutes. Afterwards, each leaflet was removed by tweezers and washed with ethanol using a plastic wash bottle. Particular attention was paid to trichomes and domatia during washing.

The contents of the glass jar were poured through a plastic funnel into a glass dividing funnel with a Teflon® stopcock. The empty glass jar and the plastic funnel were washed with 85% ethanol, which was poured into the dividing funnel immediately. All invertebrates settled at the bottom of the dividing funnel within 5 min. Then, the Teflon® stopcock was opened, and approximately 50 mL of ethanol sample containing all invertebrates was poured off into a small glass vial with a plastic plug. The material was stored in this vial until microscope slide preparation and identification.

Each vial with preserved mites was placed in a paper holder to prevent spillage, and the ethanol in the vials was gradually transferred onto a watch glass using a plastic Pasteur pipette. The sample on the watch glass was inspected using a dissection microscope (Technival, Carl Zeiss, Jena, Germany). All mites were removed with a wire loop and mounted on temporal microscope slides in lactic acid. The mites were identified using a Peraval Interphaco microscope (Carl Zeiss, Jena, Germany) and the morphological identification keys [21–25].

*2.4. Data Presentation and Statistical Analysis*

The coefficient of constancy (*C*) [26] was used to indicate the frequency of different species in the studied localities:

$$C(\%) = \frac{N_a}{N} \times 100 \tag{1}$$

where $N_a$ is the number of samples with species *a*, and *N* is the total number of samples. The species were classified as accidental (*C* < 25%), accessory (*C* = 25%–50%), constant (*C* = 50%–75%), or euconstant (*C* > 75%) [26].

The abundance of phytoseiid mites was expressed as the number of mites per compound *A. hippocastanum* leaf and analyzed by a generalized linear model using a Poisson distribution and log-link function. The following predictors were used: percentage of greenery area, intensity of greenery care (mowing/raking, ranging from 0 to 3), distance to water body, density and age of horse chestnut trees, temperature, precipitations, and overall air pollution index. The latter was calculated as a sum of indexes of individual pollutants:

$$I_t = \sum_i^n I_i \tag{2}$$

where $I_i$ is the air pollution index of pollutant *i* calculated as:

$$I_i = \frac{C_i}{Co_i} \tag{3}$$

where $C_i$ and $Co_i$ are the monitoring value and its quality standard limit in ng/m$^3$ or μg/m$^3$, respectively [27].

The analysis was performed in SAS® Studio (Release 3.81, Enterprise Edition, SAS Institute Inc., Cary, NC, USA) using the GLM procedure (PROC GENMOD) of SAS/STAT® module [28]. Since the data represented repeated measurements and thus could not be regarded as independent, the generalized estimating equation (GEE) approach [29] was used to account for within-subject correlations, using option statement REPEATED in GENMOD procedure. *p* values < 0.05 were considered statistically significant.

## 3. Results

*3.1. Species Composition*

A total of 13,903 phytoseiid mites belonging to eight species were identified. The species composition was as follows: *E. finlandicus* 96.25% (*C* = 82.33%), *Typhlodromus* (*Typhlodromus*) *pyri* Scheuten 1.35% (*C* = 10.00%), *Amblyseius andersoni* (Chant) 0.80% (*C* = 4.50%), *Neoseiulella tiliarum* (Oudemans) 0.73% (*C* = 5.00%), *K. aberrans* 0.58% (*C* = 5.83%), *Phytoseius* (*Ph.*) *macropilis* (Banks) 0.22% (*C* = 1.00%), *Paraseiulus* (*Pa.*) *triporus*

(Chant & Yoshida-Shaul) 0.06% (*C* = 0.33%), and *Paraseiulus* (*Pa.*) *talbii* (Athias-Henriot) 0.01% (*C* = 0.08%). According to the coefficient of constancy, all species can be considered accidental except *E. finlandicus*, which is a euconstant species.

### 3.2. Mite Abundance and Seasonal Dynamics

The mean abundance of phytoseiid mites per compound horse chestnut leaf across all sites was 2.53 ± 0.29 (*n* = 240), 10.40 ± 0.51 (*n* = 240), 23.54 ± 0.91 (*n* = 240), 11.59 ± 0.49 (*n* = 240), and 9.27 ± 0.49 (*n* = 240) in May, June, July, August, and September, respectively. The overall annual average abundance was 11.47 ± 0.32 mites per leaf (*n* = 1200). With the exception of Stromovka park, mite abundance culminated in July (Figure 3) and thus correlated with the mean temperature. The highest mite density was observed in the Vltava estate site, while the lowest was in Stromovka park. The GLM analysis revealed that differences among sampling sites can be associated with all considered predictor variables with the exception of precipitations (Table 3). While the effects of greenery area, care intensity, distance to water, tree density, and age on mite abundance were negative, the statistical analysis showed that the mite abundance was positively related to temperature and air pollution index. Hence, e.g., Třebotovice and Kaliště village with the lowest pollution reported (Table S2) was inhabited by a lower density of Phytoseiidae throughout the season compared to the heavily polluted Nádražní street (Figure 3).

**Table 3.** Generalized linear model (GLM) with Poisson distribution and log-link function showing the relations between mite abundance and predicting variables.

| Parameter | Estimate | Standard Error | 95% Confidence Limits | | Z | *p* |
|---|---|---|---|---|---|---|
| Intercept | −4.9278 | 0.5132 | −5.9336 | −3.9220 | −9.60 | <0.0001 |
| Greenery area | −0.0462 | 0.0026 | −0.0512 | −0.0411 | −17.93 | <0.0001 |
| Care intensity | −0.1688 | 0.0208 | −0.2096 | −0.1280 | −8.11 | <0.0001 |
| Distance to water | −0.0002 | 0.0001 | −0.0003 | −0.0000 | −2.26 | 00.0239 |
| Tree density | −0.5152 | 0.0160 | −0.5465 | −0.4838 | −32.20 | <0.0001 |
| Tree age | −0.0317 | 0.0030 | −0.0375 | −0.0258 | −10.64 | <0.0001 |
| Temperature | 0.1941 | 0.0183 | 0.1583 | 0.2300 | 10.62 | <0.0001 |
| Precipitations | −0.0009 | 0.0014 | −0.0037 | 0.0019 | −0.63 | 00.5257 |
| Air pollution index | 1.5798 | 0.0978 | 1.3881 | 1.7716 | 16.15 | <0.0001 |

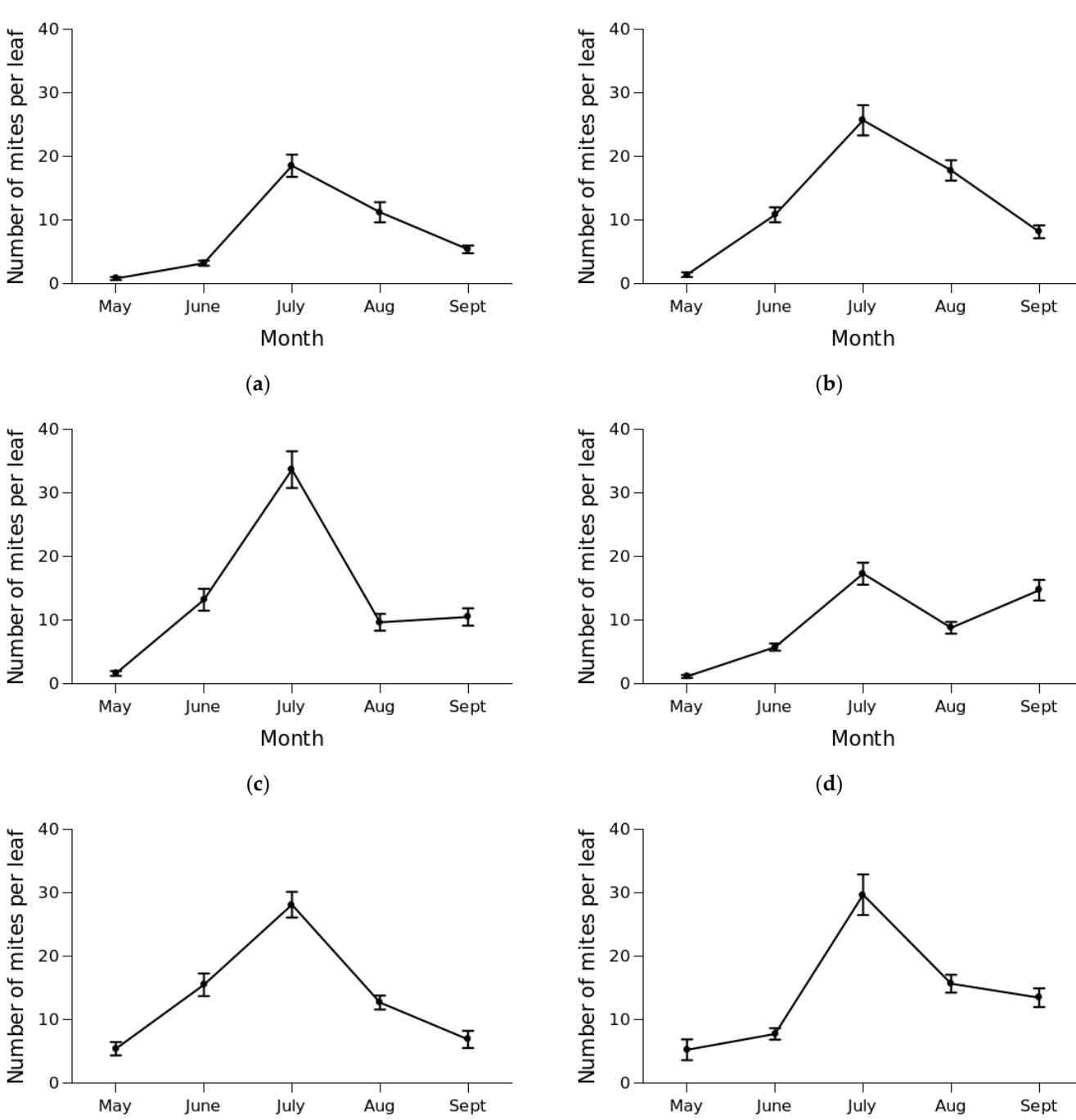

**Figure 3.** *Cont.*

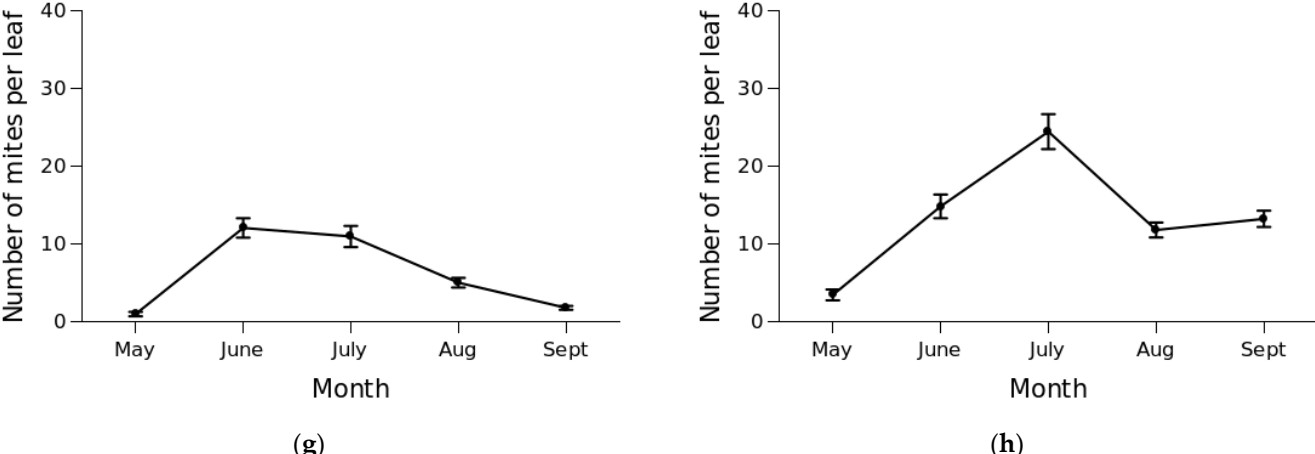

(**g**)                           (**h**)

**Figure 3.** The mean abundance of phytoseiid mites per *Aesculus hyppocastanum* compound leaf during vegetation period at eight sampling sites in České Budějovice. (**a**) City center; (**b**) Šumava and Máj estate; (**c**) Vltava estate; (**d**) Třebotovice and Kaliště village; (**e**) Rožnov estate; (**f**) Pražské předměstí estate; (**g**) Stromovka park; (**h**) Nádražní street. Vertical bars indicate standard error of the mean (*n* = 30).

## 4. Discussion

The Phytoseiidae family represents a very important group of predatory mites that inhabit, in addition to herbaceous plants, many species of deciduous trees and shrubs [9,10,30–42]. Six species of phytoseiids were identified on horse chestnut in city parks in Prague, Czech Republic, by Kabíček and Řeháková [12]: *E. finlandicus*, *Galendromus longipilus* (Nesbitt), *K. aberrans*, *Neoseiulella aceri* (Collyer), *N. tiliarum*, and *T. (T.) pyri*. The species richness, however, varied among the investigated sites, from one to four species.

The species composition of the present study confirmed the presence of these species, except *G. longipilus* and *N. aceri*, which were not found in any sample in České Budějovice. However, we found *A. andersoni*, *Ph. macropilis*, *Pa. triporus*, and *Pa. talbii*. Thus, eight species were found on *A. hippocastanum* in total. Six of them had also been found on horse chestnut in Hungary [43,44], while *Pa. talbii* and *Ph. macropilis*, which were identified in the present study, have not yet been reported on horse chestnut leaves. In Greece, where horse chestnut is an autochthonous tree, only four phytoseiid species were found, namely, *E. finlandicus*, *K. aberrans*, *Pa. talbii*, and *T. (T.) pyri* [11].

The most abundant species found in the present study was *E. finlandicus*, representing approximately 96% of all phytoseiid mites. This confirms that *E. finlandicus* is predominant in species complexes of phytoseiid mites in Europe on horse chestnut [9,12,38]. In Greece, however, the percentage of *E. finlandicus* on *A. hippocastanum* was found to be only 48%, and the second most dominant species, representing approximately 43% of all determined species of Phytoseiidae, was *K. aberrans* [11]. *Euseius finlandicus* is also the predominant species on other deciduous trees [9,10,32,35,36,38,45]. For example, in a survey by Kabíček and Povondrová [35], *E. finlandicus* represented more than 95% of all phytoseiid mites identified in leaf samples collected from various deciduous trees in a park in Prague.

The results of the present study revealed that one *A. hippocastanum* leaf can host 2.5–23.5 phytoseiid mites on average. These abundance values agree with those of previous studies conducted in Finland [9,10] and the Czech Republic [12]. The latter authors reported the highest population density of phytoseiids on horse chestnut trees in Prague to be 3.3 mites per leaflet on average [12]. Because *A. hippocastanum* compound leaves usually have five to seven leaflets [1], the above value could correspond to 19.8 mites per average compound leaf. This result matches that of our third sampling in July. In contrast, the density of phytoseiid mites in Greece was lower, with 4.2 mites per compound leaf on average [11].

Relatively high abundance of Phytoseiidae on horse chestnut compared to other deciduous tree species [32,33,35] confirms that *A. hippocastanum* is a favorable host plant species for phytoseiid generalists in urban environments. We would expect that it is due to the presence of prey availability. Although phytophagous mites are a common food source for most Phytoseiidae [31], we found only a few phytophagous mites in our samples. Among the mites infesting the horse chestnut tree are *Eotetranychus pruni* (Oudemans) (Acari: Tetranychidae) [46], *Aculus hippocastani* (Fockeu), and *Shevtchenkella carinatus* (Nalepa) (Acari: Eriophyidae) [47]. However, according to Tuovinen and Rokx [9], prey density does not seem to have any significant effect on the presence and density of *E. finlandicus* or *P. macropilis*. The number of phytoseiids on leaves is influenced by leaf surface characteristics rather than by food availability [48]. Horse chestnut trees have few glandular trichomes with a mean height of 84 μm located only on the midrib surface on the adaxial epidermis; nonglandular trichomes with lengths ranging from 116–436 μm were observed on the lower leaf surface, where they were located on the midrib and lateral veins as well as in the vein axils [49]. The density of leaf trichomes was reported to be 9.96/mm$^2$, which is relatively high compared to that in other tree species [50]. The relatively high phytoseiid density on *A. hippocastanum* can thus be explained by the favorable micromorphology of its leaves. A positive effect of leaf trichomes and domatia occurrence on the abundance of predatory mites has been well documented in many studies [32,33,48,51–61]. For example, no mites were found on tree species with glabrous leaves such as *Betula pendula* Ehrh. or *Populus tremula* L. with nonraised veins and no domatia [32,33]. While domatia mainly provide phytoseiid mites with shelter and act as protection from either natural enemies or abiotic stress [53], leaf pubescence also increases the capture and retention of pollen and fungal spores that serve as alternative foods [62]. We often observed many pollen grains on *A. hippocastanum* leaves. Kugler [63] estimated that this tree species itself can produce 42 million pollen grains from a single inflorescence. The *A. hippocastanum* pollen was also found to have a very high nutritional quality for phytoseiid mites [64,65]. Because the highest pollen concentration occurs during May [66], its availability probably facilitated the significant increase in phytoseiid mite density in June and July observed in our study. The other food resources such as extrafloral nectar or fungi are also considered to be important for generalist phytoseiid mites [67,68], and their role in nutrition of mites inhabiting horse chestnut needs to be investigated, too.

The different locality conditions may determine the variability of phytoseiid population densities on horse chestnut. Results of the present study showed that mite abundance can be affected by various factors. It is interesting that air pollution had a positive effect on density of phytoseiids. Air pollution of České Budějovice is relatively low as only benzo(a)pyrene concentration exceeded the limit (Table S2). The highest overall pollution index was found at the Nádražní street site, which is a narrow green belt between a busy road and railway line. Horse chestnut trees are thus likely to accumulate heavy metals in this site [69]. Some plants use this accumulation as a defense against herbivorous arthropods [70], and therefore a negative effect on predatory mites could be expected. Seniczak et al. reported that a high concentration of heavy metals was harmful to mites [16]. On the other hand, deposition of atmospheric nitrogen originating from vehicle exhaust can increase herbivore populations through effects on host quality [71]. Similarly, the use of de-icing salts has been implicated in the increased nutritive value of leaves on trees in street habitats and, in turn, in elevated populations of spider mites [72]. Moreover, Nádražní street is the only site without mowing and removing of leaf litter, measures which are recommended to control *C. ohridella* [73]. The highest greenery care is performed in the City center site resulting in low damage inflicted by *C. ohridella* [7,17]. However, this greenery open space maintenance was found to have a negative impact on abundance of hymenopteran parasioids and parasitism rate [15,17], so it could also negatively affect density of Phytoseiidae. Indeed, the abundance of mites in the City center site was lower than that in Nádražní street.

## 5. Conclusions

The present study showed that the horse chestnut tree grown in urban environment is a favorable host tree for phytoseiid mites. More than 90% of specimens collected were identified as *E. finlandicus*. Two species, *Pa. talbii* and *Ph. macropilis,* were recorded on horse chestnut for the first time. The other five species found were *A. andersoni*, *K. aberrans*, *N. tiliarum*, *Pa. triporus,* and *T. (T.) pyri*. A high abundance of Phytoseiidae confirmed that *A. hippocastanum* can serve as a good host plant of these predatory mites and should be planted in proximity to vineyards, orchards, or other crops for better control of spider mites and eriophyiid mites. The mite density varied significantly among sampling sites, which can be attributed to different site conditions. Whether air pollution has a positive effect, while removing leaf litter and mowing have a negative effect, on conservation of predatory mites will require more investigation.

**Supplementary Materials:** The following supporting information can be downloaded at: https://www.mdpi.com/article/10.3390/f14050942/s1, Table S1: Concentrations of air pollutants (weight per m$^3$ of air) in *Aesculus hippocastanum* sampling sites. Symbols in brackets indicate weight units. Data representing five-year averages of 2009–2013 were obtained from Geographical Information System of South Bohemia [19]; Table S2: Air pollution index for individual atmospheric pollutants and overall index of air pollution for *Aesculus hippocastanum* sampling sites. Individual indexes were calculated from measurements shown in Table S1 and concentration limits indicated in brackets using Equation (2). The concentration limits are from Geographical Information System of South Bohemia [19].

**Author Contributions:** Conceptualization, R.Z. and M.K.; methodology, R.Z. and M.K.; data curation, M.K.; writing—original draft preparation, M.K.; writing—review and editing, M.K. and R.Z.; visualization, M.K and R.Z.; supervision, R.Z. All authors have read and agreed to the published version of the manuscript.

**Funding:** This research was funded by the Czech Academy of Sciences, grant number RVO: 60077344.

**Data Availability Statement:** The data presented in this study are available on request from the corresponding author.

**Acknowledgments:** The authors thank Georgios Th. Papadoulis for his help with the identification of phytoseiid mites, Jiunn Luh Tan for his advice on data analysis, Marco del Olmo Linares for drawing graphs, and Barbora Kozelková for her technical assistance.

**Conflicts of Interest:** The authors declare no conflict of interest. The funders had no role in the design of the study; in the collection, analyses, or interpretation of data; in the writing of the manuscript; or in the decision to publish the results.

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
