# Peer review of "Species Composition and Seasonal Abundance of Predatory Mites (Acari: Phytoseiidae) Inhabiting Aesculus hippocastanum (Sapindaceae)"

_forests, doi:10.3390/f14050942_

Round 1

Reviewer 1 Report

Dear Authors

Please see my comments and suggestions in the pdf file. As according to me, you should confirm the species identification of macropilis. As you know, Phytoseiulus macropilis is taxonomically and morphologically (living form) close to a well-known tetranyhid predator, Phytoseiulus persimilis. However, I assume that your specimens are identical to Phytoseius macropilis, a common species in Europe. Please confirm this assumption and correct your manuscript accordingly. Hope this will help to improve your contribution and you will find my comments constructive. 

Kind regards

Author Response

Dear Authors

Please see my comments and suggestions in the pdf file. As according to me, you should confirm the species identification of macropilis. As you know, Phytoseiulus macropilis is taxonomically and morphologically (living form) close to a well-known tetranyhid predator, Phytoseiulus persimilis. However, I assume that your specimens are identical to Phytoseius macropilis, a common species in Europe. Please confirm this assumption and correct your manuscript accordingly. Hope this will help to improve your contribution and you will find my comments constructive.

Response: Dear Reviewer, We appreciate very much finding our mistake in writing genera name of P. macropilis mite. Indeed, it was identified as Phytoseius, not Phytoseiulus. Sorry for the typos, we corrected the names. Resposes to the comments and suggestions in annotated pdf file are written into the uploaded file (Please see the attachment).

Reviewer 2 Report

I have suggested some changes in the attached PDF, which are self-explanatory.

Please consider the following points:

A couple of more introductory sentences.

Can you justify only a one-year research study to draw a solid conclusion? The other issue is the validity of around 10-year-old data. Why the data was just sitting so long? Is it worth publishing a one-year classic data that is ten years old?

Quite a short Introduction. Must include other classical studies. 

What was the significance of getting air pollution data? 

The site map should be included.

Does 30 leaf sample represent the population? How did you decide that 30-leaf sampling represents the population? Did you collect leaves only at a 2.5 m high canopy? or it was 2.5 m and above (up to plant height). It may be depending on pest behavior to prefer older leave, etc. 

How can one leaf (which is a compound leaf) represent more than 1000 compound leaves of a tree?

How does a variation in individual pollutants affect the variation of pest species that is based on only six months' study? How can one conclude having such limited data in hand?

By the look at Figure 1, it seems that the mites population was higher in July. However, there is a lack of environmental variables data, such as Max, Min, Avg temp, RH, Rain, Wind, and Air pressure. The variation of mites population per leaf can be compared and discussed with these environmental variables.  

When you say leaf, it is not clear that you are talking about a compound leaf or leaflet. If you mean a compound leaf, which I think you mean, then please, wherever you mentioned the leaf, change it to a compound leaf.

The statistical analysis and presentation are pretty basic. The data can be analyzed using different linear/non-linear models to make limited data more appealing.

Months are just names, the main thing is the environmental variation prevails during that specific time, which is missing in the present study.

Why averages from 2009-2013, while research data is based on only five months of one year (2013). It seems that these are the five years averages of different locations. How can one justify and compare five months of mites data with five years of air pollution data? 

Author Response

I have suggested some changes in the attached PDF, which are self-explanatory.

Response: Dear Reviewer, thank you for all suggestions. We tried to address all of them as indicated below and in the uploaded PDF file.

Please consider the following points:

A couple of more introductory sentences.

Response: Few sentences were added into Abstract and Introduction, there are also some changes proposed by Reviewer 1.

Can you justify only a one-year research study to draw a solid conclusion? The other issue is the validity of around 10-year-old data. Why the data was just sitting so long? Is it worth publishing a one-year classic data that is ten years old?

Response: We understand that field studies are better when observations are over more than just one year. Unfortunately, to process all material, mount mites and identify them took a lot of time and there was only one PhD student (first author) doing this study. Nevertheless, we would not expect any large changes between consecutive years as conditions do not change much (trees are in the city parks of České Budějovice for decades). Main message of this study is to show that A. hippocastani is good reservoir of predatory mites with single species dominating while there is high spatial variability in mite density. Indeed, repeting of this study in e.g. 10-years period would be great idea to see the effect of global warming etc., but currently there is no project nore motivated student to do this work. We still believe these results are worth publishing as they demonstrates other importance of horse chestnut in addition to ornamental function.

Quite a short Introduction. Must include other classical studies.

Response: Introduction was more ellaborated and some relevant references added.

What was the significance of getting air pollution data?

Response: We indicated in Introduction and in Discussion how pollution/accummulation of heavy metals can affect mite populations. Positive effect of pollution on phytoseiids density found in our study might be surprising and shows that further studies focused on this interaction would be needed.

The site map should be included.

Response: The map was added into Material and Methods (Figure 1).

Does 30 leaf sample represent the population? How did you decide that 30-leaf sampling represents the population? Did you collect leaves only at a 2.5 m high canopy? or it was 2.5 m and above (up to plant height). It may be depending on pest behavior to prefer older leave, etc.

How can one leaf (which is a compound leaf) represent more than 1000 compound leaves of a tree?

Response: Sample size was mostly limited by time needed to process relatively large compound leaves (each leaf had to be washed by ethanol individually) in short time. The height from which leaves were collected was the same as recommended for C. ohridella sampling. We are aware that it would be nice to have samples through canopy height but it would require a special equipment for working at height which was impossible for us.

How does a variation in individual pollutants affect the variation of pest species that is based on only six months' study? How can one conclude having such limited data in hand?

Response: We did not analyzed data using individual pollutants but overall pollution index. As regards reasons to include air pollution as explanatory variable please see above.

By the look at Figure 1, it seems that the mites population was higher in July. However, there is a lack of environmental variables data, such as Max, Min, Avg temp, RH, Rain, Wind, and Air pressure. The variation of mites population per leaf can be compared and discussed with these environmental variables.

Response: Thank you for suggestion to include weather data. Including them in the statistical analysis helped to explain seasonal patern.

When you say leaf, it is not clear that you are talking about a compound leaf or leaflet. If you mean a compound leaf, which I think you mean, then please, wherever you mentioned the leaf, change it to a compound leaf.

Response: All terms “leaf” was changed to “compound leaf”.

The statistical analysis and presentation are pretty basic. The data can be analyzed using different linear/non-linear models to make limited data more appealing.

Months are just names, the main thing is the environmental variation prevails during that specific time, which is missing in the present study.

Response: As written above, we included weather data in the model.

Why averages from 2009-2013, while research data is based on only five months of one year (2013). It seems that these are the five years averages of different locations. How can one justify and compare five months of mites data with five years of air pollution data?

Response: We took 5-years averages because it better reflects potential accummulation of pollutants in trees and a long-term effect on tree vigour rather than single measurement of pollution in year of mite sampling.

Reviewer 3 Report

The authors carried out a classical study of the species composition of mites on horse chestnut, using classical methods and approaches. I have no significant comments on the manuscript aside this one, that I ask the authors to answer the question.

Why did the data obtained in 2013 not glare for about 10 years? Why is there no data for 2014, 2015.

These data would greatly embellish the scientific component of the manuscript.

Also,
there are citations of works in Ukrainian in the manuscript. All of them have abstracts in English, so they can be presented according to the rules, thus readers can find them much faster.

Author Response

The authors carried out a classical study of the species composition of mites on horse chestnut, using classical methods and approaches. I have no significant comments on the manuscript aside this one, that I ask the authors to answer the question.

Why did the data obtained in 2013 not glare for about 10 years? Why is there no data for 2014, 2015.

These data would greatly embellish the scientific component of the manuscript.

Response: Dear Reviewer, That is definitely a good question and we understand that field studies are better when observations are over more than just one year. Unfortunately, to process all material, mount mites and identify them took a lot of time and there was only one PhD student (first author) doing this study. Nevertheless, we would not expect any large changes between years and think new study would be good to repeat every 10 years to see e.g. the effect of global warming. However, currently there is no grant project or motivated student to do this work.

Also, there are citations of works in Ukrainian in the manuscript. All of them have abstracts in English, so they can be presented according to the rules, thus readers can find them much faster.

Response: Title of these papers were changed to English.

Round 2

Reviewer 1 Report

Well done!

Author Response

Dear Reviewer,

Thank you for a positive review. We are glad you like the revised version of our manuscript.

Best regards,

Rostislav Zemek

Reviewer 2 Report

Please check the scientific names as highlighted in the attached PDF files.

Author Response

Response: Dear Reviewer, Thank you for all suggestions. We checked the species names and added (Ph.) and (Pa.) abbretiviation when genera names of Phytoseius and Paraseiulus occurred for the first time (First paragraph of Results, now at page 6) as to us seems to be more convenient for readers to avoid confusion. On page 1 and 2 we, however, did not change Aesculus hippocastanum to abbreviated version because it is recommended to write species name in full when it starts the sentence as it looks better. As regards Figure 3 we modified it as suggested. We also improved resolution of Figure 2.

Reviewer 3 Report

I have read the revised manuscript of the article. The authors responded to my recommendations and questions. There are no comments on the second version, the work can be published in present form.

Author Response

Dear Reviewer, Thank you very much for a positive review.